# The membrane associated accessory protein is an adeno-associated viral egress factor

Zachary C. Elmore[1], L. Patrick Havlik[1], Daniel K. Oh[1], Leif Anderson [2], George Daaboul[2], Garth W. Devlin[1], Heather A. Vincent[1] & Aravind Asokan [1,3,4,5✉]

Adeno-associated viruses (AAV) rely on helper viruses to transition from latency to lytic infection. Some AAV serotypes are secreted in a pre-lytic manner as free or extracellular vesicle (EV)-associated particles, although mechanisms underlying such are unknown. Here, we discover that the membrane-associated accessory protein (MAAP), expressed from a frameshifted open reading frame in the AAV *cap* gene, is a novel viral egress factor. MAAP contains a highly conserved, cationic amphipathic domain critical for AAV secretion. Wild type or recombinant AAV with a mutated MAAP start site (MAAPΔ) show markedly attenuated secretion and correspondingly, increased intracellular retention. Trans-complementation with MAAP restored secretion of multiple AAV/MAAPΔ serotypes. Further, multiple processing and analytical methods corroborate that one plausible mechanism by which MAAP promotes viral egress is through AAV/EV association. In addition to characterizing a novel viral egress factor, we highlight a prospective engineering platform to modulate secretion of AAV vectors or other EV-associated cargo.

[1] Department of Surgery, Duke University School of Medicine, Durham, NC, USA. [2] Nanoview Biosciences, Boston, MA, USA. [3] Department of Molecular Genetics & Microbiology, Duke University School of Medicine, Durham, NC, USA. [4] Department of Biomedical Engineering, Duke University, Durham, NC, USA. [5] Duke Regeneration Center, Duke University, Durham, NC, USA. ✉email: aravind.asokan@duke.edu

Adeno-associated virus (AAV) is a non-enveloped, single-stranded DNA virus belonging to the Dependoparvovirus genus within the Parvoviridae family[1]. Upon co-infection with a helper such as Adenovirus, Herpesvirus or Papillomavirus, AAV undergoes a transition from latent to lytic life cycle, hijacking the host cell machinery[1]. The AAV capsid consists of 60 capsid monomers of VP1, VP2, and VP3 at a ratio of 1:1:10 that packages a 4.7-kb single-stranded genome[2]. The AAV genome encodes replication (rep), capsid (cap), and assembly-activating protein (AAP) open reading frames (ORFs) flanked by inverted terminal repeats (ITRs), which are the sole requirements for genome packaging[3–5]. As such, the majority of the genome can be replaced by exogenous DNA sequences and packaged inside the AAV capsid to create a recombinant vector both in vitro and in vivo[6,7]. Some recombinant AAV serotypes appear to be secreted into cell culture media prior to lysis, albeit with variable efficiency[8–10]. Exactly how AAV exits the cell upon transitioning into this phase of replication remains unclear. Unlike other autonomous parvoviruses that undergo a lytic cycle, wild type AAV does not induce marked cytopathic effects (CPE) and therefore, cellular egress of virions is thought to be primarily driven by overexpression of Adenoviral or Herpesvirus proteins[11,12].

Multiple studies have demonstrated that a significant fraction of recombinant AAV associate with extracellular vesicles (EVs) and are also released as free particles into the supernatant fraction of the cell culture media[13]. It is well known that cells shed a variety of membrane-bound vesicles varying in size from 20 nm to 1 μm in diameter, which have been termed exosomes, microparticles or microvesicles (collectively referred to here as extracellular vesicles or EVs). Such EVs can package different macromolecules including proteins, nucleic acids and viruses, thereby making them an attractive therapeutic platform[14]. Despite being non-enveloped viruses, recombinant AAV capsids associated with EVs can enable efficient gene transfer to the retina, the nervous system, the inner ear[15–17] and appear shielded from anti-AAV neutralizing antibodies[17]. However, the mechanism(s) by which AAV exits the cell or associates with EVs remain to be determined.

Recent work has revealed a novel +1 frameshifted open reading frame (ORF) in the VP1 region of the AAV cap gene that mediates expression of the membrane-associated accessory protein (MAAP) (Fig. 1A), which was postulated to limit AAV production through competitive exclusion[18]. Here, we assign a novel function to MAAP in promoting AAV egress from host cells. These events are mediated, at least in part, due to MAAP-enabled association of AAV with EVs. Specifically, we dissect the secondary structure elements in MAAP that contribute to this secretory function. Further, we demonstrate that MAAP trans-complementation can not only rescue but significantly alter the kinetics of viral secretion of multiple AAV serotypes. Further, independent of AAV infection, MAAP alone appears to associate significantly with EVs. Our findings highlight a viral egress mechanism evolved by AAV as well as highlight the potential to exploit this novel protein for loading therapeutic cargo including AAV vectors into EVs and AAV vector production in general.

## Results

**MAAP is a conserved AAV *cap* encoded protein with a predicted C-terminal membrane anchoring domain.** Confirming that MAAP is a novel viral-encoded protein of unknown function, a pBLAST search of multiple AAV cap gene derived MAAP sequences on the National Center for Biotechnology Information (NCBI) website did not return any proteins with significant homology[18]. Amino acid sequence alignment of MAAPs derived from different AAV serotypes revealed conserved N- and C-terminal regions containing hydrophobic and basic amino acid residues interconnected by a threonine/serine (T/S) rich region (Fig. 1B). MAAP 3D structures generated using TrRosetta deep-learning-based modeling[19] predicted the following, (i) a conserved N-terminal hydrophobic motif with both alpha helical and beta strand secondary structure elements; (ii) four T/S rich sequence clusters spanning 7–17 residues in length with last two being separated by a smaller alpha helical interspersed with basic residues and (iii) a C-terminal domain defined by another hydrophobic alpha helical motif merging into a cluster of arginine/lysine (R/K) residues (Fig. 1B, C). Importantly, secondary structure software analysis strongly predicts that the C-terminal domain constitutes a putative membrane binding, cationic amphipathic peptide (residues 96–114). It is noteworthy to mention that the secondary structure of MAAP is strikingly similar to the assembly-activating protein (AAP), which is similarly encoded downstream from a (+1) frameshifted ORF in the cap gene. When combined with phylogenetic analysis using the neighbor-joining tree method, we observed that MAAPs from AAV serotypes 1,6,8,10,11 were tightly clustered, while other sequences, in particular, MAAP2, 5 and 9 showed significant divergence from other serotypes (Fig. 1D). We then transfected plasmids encoding recombinant MAAPs derived from the VP1 sequences of AAV serotypes 1, 2, 5, 8, and 9 and fused to a C-terminal green fluorescent protein (GFP) in vitro to assess their expression (Fig. 1E) and cellular localization. Fluorescence micrographs confirmed the propensity of MAAP to associate with cell surface membranes as well as subcellular organelles as observed by the punctate patterns throughout the cell (Fig. 1F). Taken together, these data confirm that MAAP is a novel AAV protein predicted to contain a cationic amphipathic C-terminal domain potentially critical for membrane anchoring.

**MAAP is essential for extracellular secretion of wild type and recombinant AAV particles.** We then sought to determine whether MAAP played a role in the synthesis of (i) (pseudo)wild type AAV serotype 8 (i.e., wtAAV8 packaging AAV2 rep and AAV8 cap flanked by AAV2 inverted terminal repeats [ITRs]) (Fig. 2A) and (ii) recombinant AAV8 (i.e., rAAV8 packaging a chicken beta-actin promoter driven luciferase transgene flanked by AAV2 ITRs) (Fig. 2D). To ablate MAAP expression, we mutated the CTG start codon in the MAAP alternative open reading frame (ORF) without affecting the VP1 ORF in both wtAAV8 and rAAV8 plasmids. Culture media and cell pellets were harvested following co-transfection with an Adenovirus helper plasmid (and an additional ITR flanked luciferase encoding transgene cassette in case of rAAV) on days 3 and 5. Strikingly, quantitative PCR of viral genomes revealed a significantly higher (~1 log) amount of extracellular wtAAV8 particles in contrast to MAAPΔ particles recovered from media on day 3 (Fig. 2B). Further, we observed delayed secretion in case of MAAPΔ particles, which were equally apportioned between extracellular and cell lysate fractions on day 5. Although statistically significant, overall viral titers on day 5 were only minimally altered. Of the total virus produced, nearly 70% of wtAAV8 particles were secreted by day 3, while MAAPΔ particles recovered in the extracellular fraction comprised <10% of total (Fig. 2C).

A similar trend was observed in case of rAAV8 particles, with a 4–5-fold higher recovery from media over cell lysate and delayed secretion in case of MAAPΔ particles (Fig. 2E). Of the total virus produced, ~60% of rAAV8 particles were secreted by day 3 in contrast to <10% of MAAPΔ particles (Fig. 2F). Further evaluation of AAV capsid proteins – VP1,2 and 3 by western

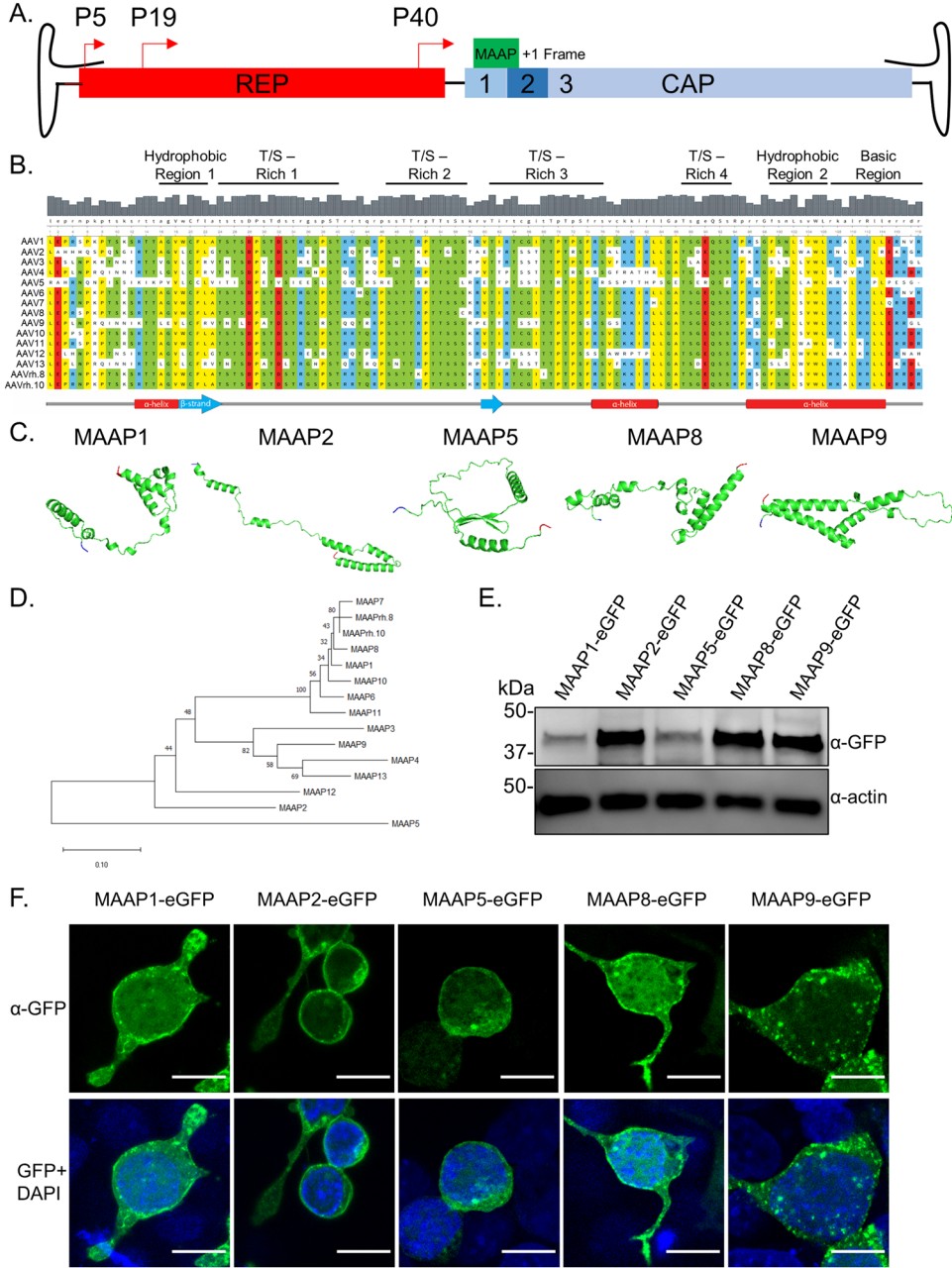

**Fig. 1 MAAP is a unique virally encoded protein with an amphipathic, cationic membrane anchoring domain. A** WT AAV genome showing *Rep* and *Cap* genes with MAAP encoded in a +1 open reading frame in the VP1 region. **B** Sequence alignment of the MAAPs from AAV serotypes 1 to 13 along with AAVrh.8 and AAVrh.10. An annotated multiple-sequence alignment of the AAP sequences of 15 AAV serotypes is shown. Coloring reflects the physicochemical properties of the residues (Yellow = hydrophobic, Green = polar, Blue = basic, Red = acidic). Regions of interest are annotated above the alignment. Predicted secondary structural (SS) elements (strand, helix) for the sequence and amino acid numbering are displayed below the alignments. **C** Structural models of MAAP1, MAAP2, MAAP5, MAAP8, and MAAP9 generated using Robetta Structural Prediction Server. Residues highlighted in blue indicate N-terminus and residues in red indicate C-terminus. **D** Neighbor-joining phylogeny of MAAP amino acid sequences from AAV serotypes 1 to 13, MAAPrh.8 and MAAPrh.10. MAAP amino acid sequences were aligned with ClustalW, the phylogeny was generated using a neighbor-joining algorithm, and a Poisson correction was used to calculate amino acid distances, represented as units of the number of amino acid substitutions per site. The tree is drawn to scale, with branch lengths in the same units as those of the evolutionary distances used to infer the tree. Bootstrap values were calculated with 1000 replicates, and the percentage of replicate trees in which the associated taxa clustered together are shown next to the branches. **E** Anti-GFP immunoblot of whole-cell extracts prepared from HEK293 cells expressing indicated GFP tagged constructs. Anti-actin immunoblot served as loading control. Immunoblot is a representative image of four independent experiments. **F** Confocal images of HEK293 cells overexpressing eGFP tagged MAAP constructs. Scale bar = 10 μM. Micrographs are representative images of three independent experiments.

blot confirmed these results with undetectable to relatively lower levels in the extracellular fraction on days 3 and 5, respectively, and correspondingly high(er) cellular retention in case of MAAPΔ particles (Fig. 2G, H). No differences were observed when comparing the transduction efficiency of rAAV8 and MAAPΔ particles in vitro (Fig. 2I). Furthermore, MAAPΔ recombinant virus showed similar VP1, VP2, and VP3 expression ratios and overall virus morphology compared to rAAV8

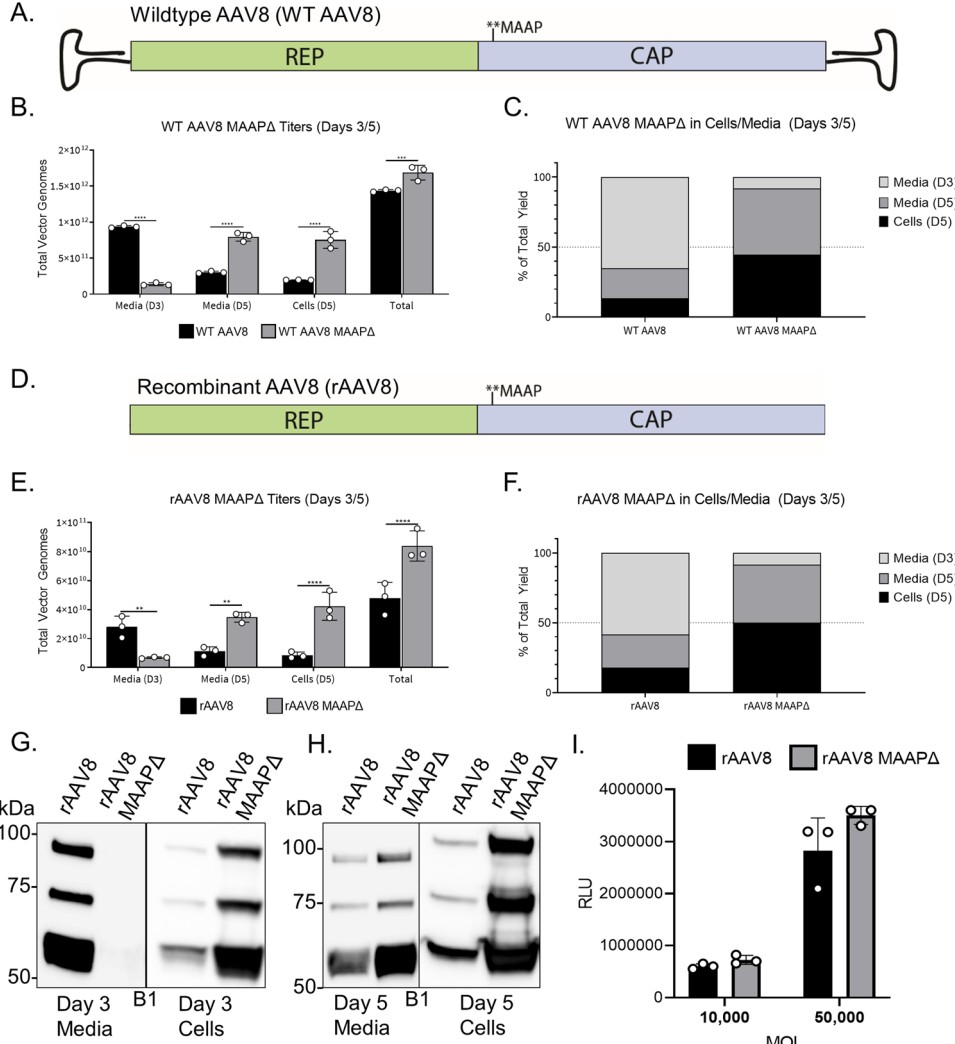

**Fig. 2 Ablation of MAAP expression results in a significant delay in the extracellular secretion of wildtype and recombinant AAV8 particles. A** Schematic of WT AAV8 MAAPΔ mutant. **B** AAV8 ssCBA-Luc vectors produced with WT Cap or MAAPΔ Cap. Total vector genomes collected from the cells and media and the proportion of virus found in each media harvest or associated with the cells **C** are shown. Data are presented as mean values ± SD. Significance was determined by two-way ANOVA, with Sidak's post-test. (Media (D3) c Media (D5) ****$p < 0.0001$; Cells (D5) ****$p < 0.0001$, Total ***$p = 0.0004$). **D** Schematic of rAAV8 MAAPΔ mutant. **E** AAV8 ssCBA-Luc vectors produced with recombinant Cap or MAAPΔ Cap. Total vector genomes collected from the cells and media and the proportion of virus found in each media harvest or associated with the cells (**F**) are shown. Data are presented as mean values ± SD. Significance was determined by two-way ANOVA, with Sidak's post-test. (Media (D3) **$p = 0.0081$; Media (D5) **$p = 0.0036$; Cells (D5) ****$p < 0.0001$, Total ****$p < 0.0001$). Recombinant AAV8 and AAV8 MAAPΔ viruses were analyzed from the media and pellet of HEK293 producing cells at days 3 (**G**) and 5 (**H**) post transfection. Capsid proteins were analyzed by SDS-PAGE under reducing conditions and probed with a capsid (B1)-specific antibody. Immunoblots are representative images of two independent experiments. **I** Luciferase assay analyzing transduction of HEK293 cells by AAV8 and AAV8 MAAPΔ mutant virus at MOIs of 10,000 and 50,000 vg/cell. Each bar is a representation of three experiments that are biological replicates.

(Supplementary Fig. 1). Taken together, these results suggest that encoding MAAP from the alternative ORF in VP1 is essential for efficient cellular egress of AAV particles. Another interesting observation is the relatively low recovery of recombinant AAV serotype 9 (rAAV9) (~15%) and MAAPΔ particles (~5%) from media on day 3 (Supplementary Fig. 2). In contrast to rAAV8, rAAV9 particles appear to display delayed secretion as reported previously[20], with only a modest difference in cellular egress efficiency compared to MAAPΔ particles. Thus, MAAP8 has a higher propensity to promote viral egress when compared to MAAP9. When taken together with the differences in sequence homology between AAV8 and AAV9, it is tempting to speculate that the cellular egress efficiency of different AAV capsids could be determined by their cognate MAAPs.

**Helical and cationic domains are indispensable for MAAP expression and supporting AAV secretion.** We next sought to determine the regions and/or domains of MAAP necessary to promote AAV viral egress. We first inserted a 3X-FLAG tag onto the C-terminus of MAAP8 under the control of its endogenous promoter and verified its expression and function (Supplementary Fig. 3). At the primary structure level, MAAP can be separated into multiple regions including N-terminus, Linker, C-terminus, hydrophobic regions 1 and 2, threonine/serine-rich regions 1–4, (T/S), and the basic region. Taking a structure–function approach, we generated multiple MAAP8 deletion mutants to systemically dissect the regions of MAAP critical for AAV viral egress (Fig. 3A). We discovered that deletions of the N or C-terminus severely diminished MAAP

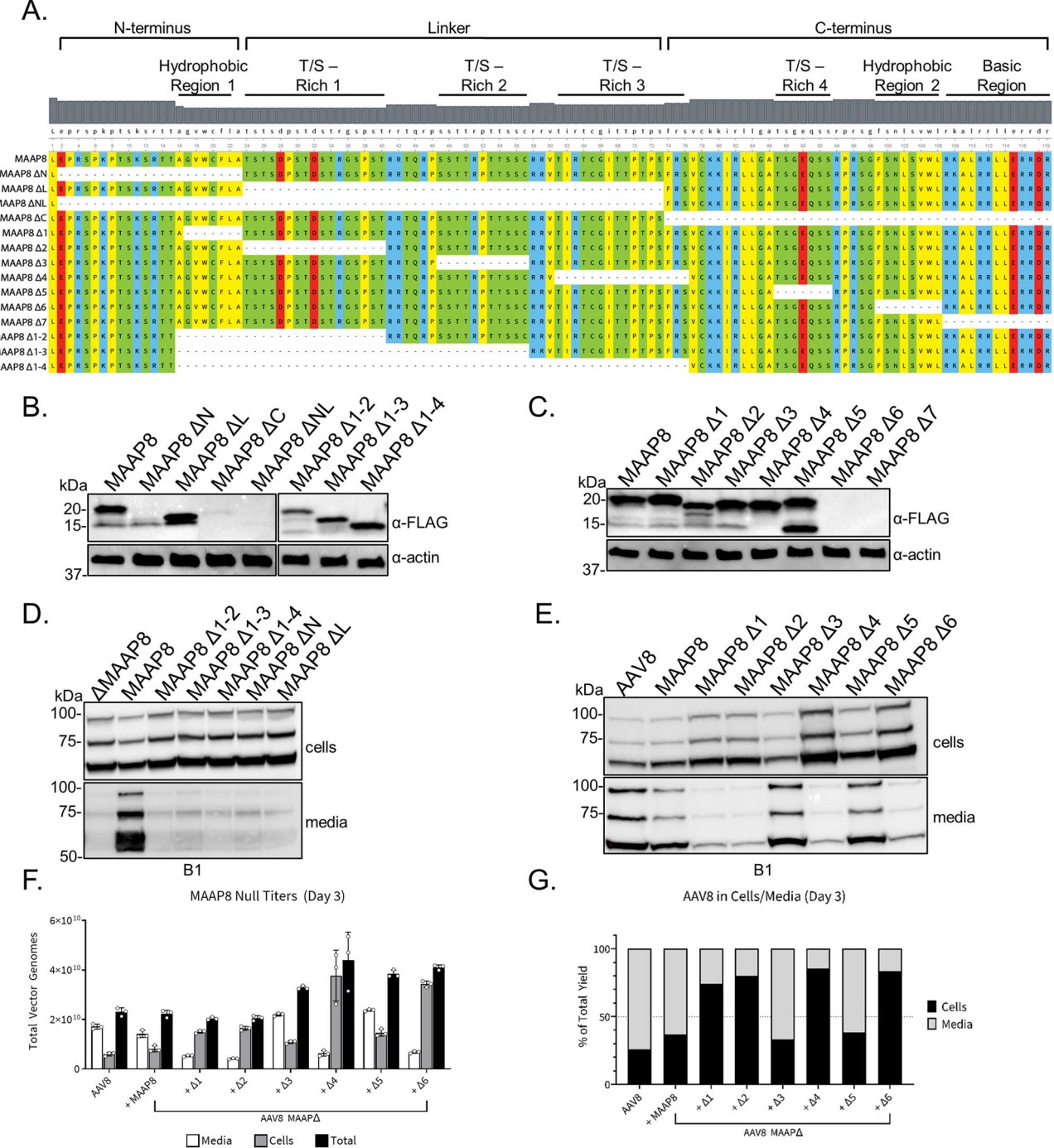

**Fig. 3 Structure–function analysis of MAAP8 reveals regions critical for expression and AAV secretion. A** Schematic of different MAAP mutants. All MAAP mutants have a 3X-FLAG tag at the C terminus. **B**, **C** Anti-FLAG immunoblot of whole-cell extracts prepared from HEK293 cells expressing indicated MAAP8-3X-FLAG tagged constructs. Anti-actin immunoblot served as loading control. Immunoblots are representative images of two independent experiments. **D**, **E** Recombinant MAAP8Δ vectors complemented *in trans* with various truncated MAAP8-3X-FLAG plasmids were analyzed from the media and pellet of HEK293 producing cells at day 3 post transfection. Capsid proteins were analyzed by SDS-PAGE under reducing conditions and probed with a capsid (B1)-specific antibody. Immunoblots are representative images of two independent experiments. Total vector genomes (**F**) and the proportion of vector found in the media and cells (**G**) 3 days post transfection. Each bar is a representation of three experiments that are biological replicates. Data are presented as mean values ± SD.

expression (Fig. 3B, C). MAAP deletions involving alpha helical domains or amphipathic domains rich in basic residues demonstrated the most deleterious effects on protein expression at steady state. Furthermore, of the MAAP8 deletion mutants that expressed, only T/S-rich region 2 and T/S-rich region 4 (MAAP8

Δ3 and MAAP8 Δ5, respectively) were found to be dispensable for MAAP function as measured through AAV secretion into the media (Fig. 3D–G). Taken together, these data identify critical secondary structure elements in MAAP that are likely essential for expression and function.

**MAAP transcomplementation promotes extracellular secretion of diverse AAV serotypes**. To determine whether MAAP expression regulates the secretion of other AAV serotypes, we mutated the CTG start codon in the MAAP ORF for rAAV serotypes 1, 2, 8, and 9, and compared viral titers in extracellular and cellular fractions at day 3 as described earlier. In parallel, we also evaluated whether MAAP transcomplementation could rescue the extracellular secretion of MAAPΔ rAAV particles. To achieve the latter, we expressed MAAP alone from the AAV helper plasmid containing *rep* and *cap* genes by mutating the start codons in the VP1,2,3 as well as AAP ORFs. Strikingly, viral titers associated with the cellular fraction were markedly increased for rAAV1, rAAV8 and rAAV9 (~4 to 7-fold), but not rAAV2 (Fig. 4A, C, E, G). In addition, overall recovered titers were increased moderately for the same serotypes (up to 2-fold). In corollary, we observed a striking impact of ablating or supplementing MAAP expression on extracellular vs cell-associated fractions of different AAV serotypes. Specifically, in case of rAAV1, this percentage was reversed from 60:40 to 20:80 upon MAAP ablation and restored to normal upon MAAP expression (Fig. 4B). A similar trend was observed for rAAV8 (~80:20 to 20:80 followed by restoration to normal upon supplementation) (Fig. 4F). Both rAAV2 and rAAV9 showed decreased secretion in general (~35:65) compared to rAAV1/8 with MAAP ablation further reducing extracellular viral titers to 15% and 20%, respectively (Fig. 4D, H). Another important observation is that MAAP8 *trans*-complementation not only fully rescued the extracellular secretion of rAAV1, rAAV2, and rAAV8 particles, but also doubled the recovery of rAAV9 MAAPΔ particles from media compared to rAAV9 particles (from 40% to 80%) (Fig. 4H). These results confirm the critical role played by MAAP in enabling extracellular secretion of AAV particles in a serotype-independent manner, albeit with different efficiencies. Further, our results demonstrate that *trans*-complementation of MAAP derived from AAV8 can not only rescue secretion of different AAV serotypes, but also potentially enhance the kinetics of secretion.

**MAAP promotes association between AAV and extracellular vesicles**. To further explore the biology of the MAAP-dependent AAV secretory process, we adopted a gradient centrifugation method to purify EVs from large volumes of cell culture supernatant[21,22]. Serum-free cell culture medium from suspension adapted HEK293 cells transfected to generate rAAV8/MAAPΔ particles (complemented in *trans* with MAAP8-HA or HA alone) was processed by successive filtration and centrifugation steps to generate a crude EV pellet that was then separated on an iodixanol gradient (Fig. 5A). The gradient was split into 18 different fractions and probed for EV markers (CD9, CD63, CD81), MAAP (HA tag) and AAV capsid protein using specific antibodies. Using sodium dodecyl sulphate–polyacrylamide gel electrophoresis (SDS-PAGE), immunoblot analysis and qPCR, we found that MAAP-associated EV fractions co-eluted with genome packaging AAV particles (Fig. 5B–D). Furthermore, overexpression of MAAP8-HA or MAAP8-GFP, but not just HA/GFP alone, was sufficient to generate a higher density MAAP-associated sub-population of EVs (Supplementary Fig. 4).

To further understand this interaction between MAAP and EVs, we utilized size exclusion chromatography (SEC) to probe different vesicular/particulate sub-populations (Fig. 5E). Serum-free cell culture medium from suspension adapted HEK293cells was processed by centrifugation and tangential flow filtration and then resolved on a CL-2B sepharose column. Of the total 30 different fractions collected and probed with EV and MAAP-HA tag-specific

antibodies, we determined that MAAP selectively induced a shift in size and was enriched in larger CD63 + vesicular fractions (Fig. 5F, G). Interestingly, analysis of vector genome titers revealed a corresponding shift in rAAV particles confirming that MAAP enables AAV association with CD63 + EVs (Fig. 5H). Taken together, these data indicate that MAAP is important in promoting the association between AAV and EVs, which likely plays a partial, yet significant role in AAV egress.

**MAAP is loaded onto the surface of extracellular vesicles**. To understand how MAAP interacts with EVs, we first carried out quantitative confocal fluorescence microscopy of cells overexpressing hemagglutinin (HA) tagged MAAP8 (confirmed by western blot in Supplementary Fig. 5). MAAP8-HA colocalized significantly more with the exosomal biogenesis pathway marker Rab11[23–25], than the late endo/lysosomal pathway marker Rab7[26] (Fig. 6A, B). Interestingly, MAAP9, which was associated earlier with slower secretion kinetics, showed a similar co-localization to both Rab7 and Rab11 subcellular compartments (Supplementary Fig. 6). These observations complement previous reports that AAV particles can associate with secreted EVs[13,15–17,27,28]. Further, our results also highlight potential differences in the structural attributes of different MAAPs that may explain the ability to exploit distinct secretory mechanisms that enables cellular egress of different AAV serotypes. Although the exact components of the vesicular secretion pathway exploited in MAAP-aided AAV egress still need to be elucidated, we carried out biochemical assays to explore potential MAAP-AAV capsid interactions. Notably, we did not find any evidence of direct interaction between MAAP and AAV capsid proteins or MAAP and AAP as determined by immunoprecipitation analysis (Supplementary Fig. 7A, B). However, it is important to note that MAAP8 fused to the BirA biotin ligase BioID2[29] was able to successfully biotinylate the AAV capsid proteins in vitro indicating that AAV and MAAP do share a proximal interaction within the cell (Supplementary Fig. 7C–F).

We then characterized EVs further using the ExoView platform (NanoView Biosciences, USA) as outlined in the methods. Briefly, the ExoView tetraspanin kit features chips with immobilized antibodies against the tetraspanins CD81, CD63 and CD9, which were used to capture EVs from media of cells expressing HA tag alone or MAAP8-HA followed by interferometric analysis and detection of EVs with fluorophore-conjugated antibodies against these tetraspanins. After EV capture and analysis, we determined that overexpression of MAAP8 does not increase the number or size of EVs in the detectable range (50–200 nm) secreted into the media (Fig. 6C, D). However, it should be noted that this assay does not detect large vesicular bodies previously recovered by SEC. Moreover, using a fluorescently labeled anti-HA tag antibody, we confirmed that MAAP8 associates definitively with captured EVs (Fig. 6E, F). Furthermore, permeabilizing captured EVs to release luminal contents (represented by Syntenin as a marker) revealed that MAAP8 was in fact associated with the outer surface of EVs (Fig. 6G, H). Taken together, these data reveal that MAAP is a virally encoded egress factor that promotes cellular exit of AAV particles at least partially through association with EVs.

## Discussion

Dissecting new aspects of AAV biology is critical for continued improvement of recombinant vectors for gene therapy applications. For instance, our lab and others previously dissected the functional attributes of the Assembly-Activating Protein (AAP), which is encoded from an alternative ORF in the AAV *cap* gene[4,5,30]. These studies provided mechanistic insight into AAV capsid assembly, in addition to highlighting capsid mutations that

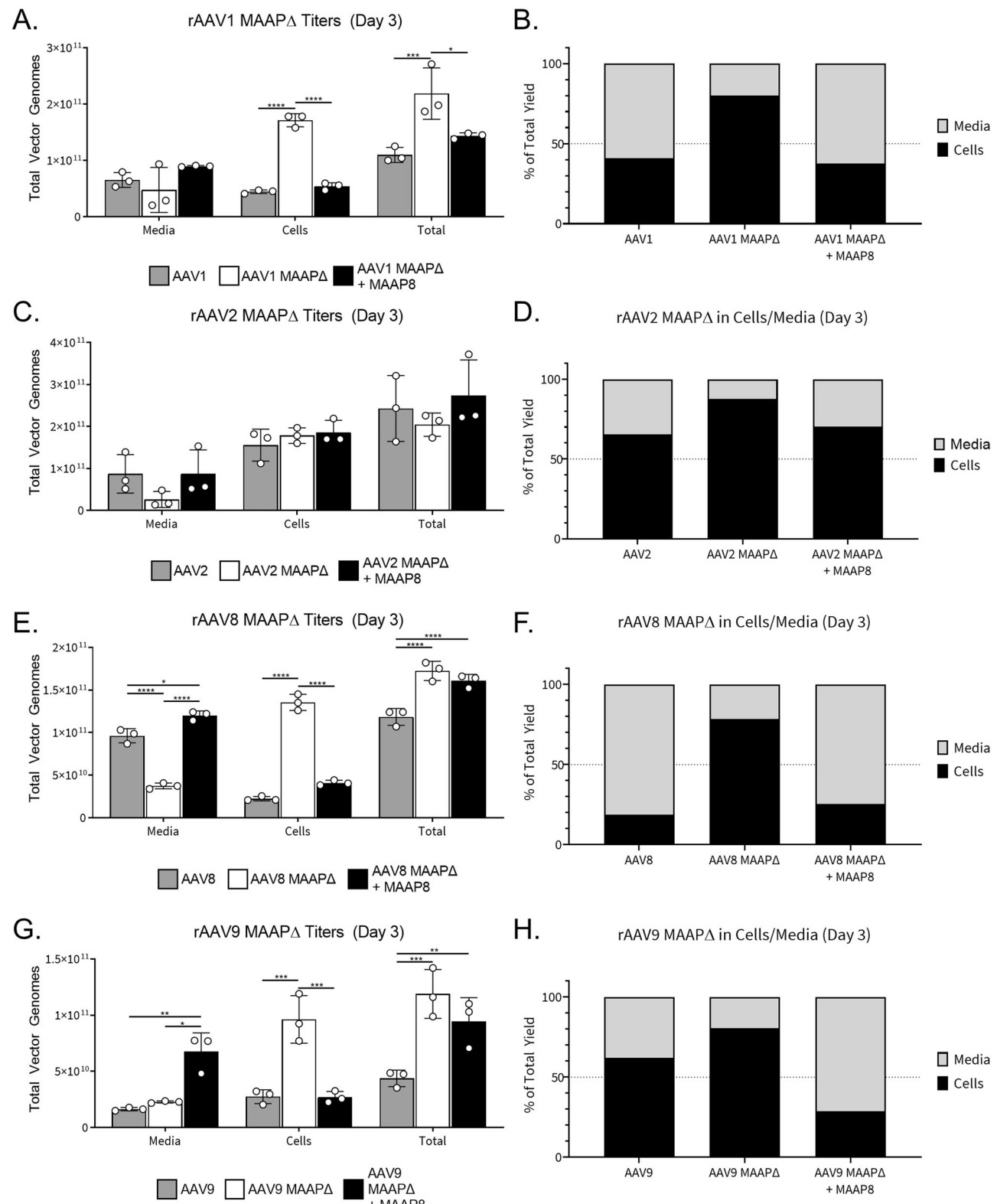

can adversely impact assembly and vector yield. Here, we provide structural and functional insights into the recently identified membrane-associated accessory protein (MAAP), which is encoded from a different, alternative ORF within the AAV *cap* gene. We report that this highly conserved protein displays critical hydrophobic and cationic domains essential for membrane anchoring and promotion of extracellular secretion of AAV particles. Although attenuation of MAAP expression does not

affect viral yield, we demonstrate that extracellular secretion of AAV particles is drastically delayed with increased cellular retention of AAV particles. Our findings have immediate implications for recombinant AAV vector production, which is currently undergoing a revolutionary manufacturing scale-up phase[7,31]. Specifically, we demonstrate that MAAP transcomplementation can significantly alter the kinetics of AAV extracellular secretion, potentially facilitating downstream purification only

**Fig. 4 Trans-complementation of MAAP8 rescues secretion defect across multiple MAAPΔ AAV serotypes.** scCBh-GFP vectors produced with WT Cap or MAAPΔ Cap for AAV1 (**A**, **B**), AAV2 (**C**, **D**), AAV8 (**E**, **F**), and AAV9 (**G**, **H**) complemented *in trans* with a VP/AAP-null AAV8 plasmid to replicate endogenous levels of MAAP expression. Total vector genomes (**A**, **C**, **E**, **G**) and the proportion of vector found in the media and cells (**B**, **D**, **F**, **H**) 3 days post transfection. Each bar is a representation of three experiments that are biological replicates. Data are presented as mean values ± SD. Significance was determined by two-way ANOVA, with Tukey's post-test. (AAV1 Cells vs AAV1 MAAPΔ Cells, ****$p < 0.0001$; AAV1 MAAPΔ Cells vs AAV1 MAAPΔ + MAAP8 Cells, ****$p < 0.0001$; AAV1 Total vs AAV1 MAAPΔ Total, ***$p = 0.0002$; AAV1 MAAPΔ Total vs AAV1 MAAPΔ + MAAP8 Total, *$p = 0.0116$; AAV8 Media vs AAV8 MAAPΔ Media, ****$p < 0.0001$; AAV8 Media vs AAV8 MAAPΔ + MAAP8 Media, *$p = 0.0211$; AAV8 MAAPΔ Media vs AAV8 MAAPΔ + MAAP8 Media, ****$p < 0.0001$; AAV8 Cells vs AAV8 MAAPΔ Cells, ****$p < 0.0001$; AAV8 MAAPΔ Cells vs AAV8 MAAPΔ + MAAP8 Cells, ****$p < 0.0001$; AAV8 Total vs AAV8 MAAPΔ Total, *****$p < 0.0001$; AAV8 Total vs AAV8 MAAPΔ + MAAP8 Total, ****$p < 0.0001$; AAV9 Media vs AAV9 MAAPΔ + MAAP8 Media, **$p = 0.0071$; AAV9 MAAPΔ Media vs AAV9 MAAPΔ + MAAP8 Media, *$p = 0.0223$; AAV9 Cells vs AAV9 MAAPΔ Cells, ***$p = 0.0003$; AAV9 MAAPΔ Cells vs AAV9 MAAPΔ + MAAP8 Cells, ***$p = 0.0003$;; AAV9 Total vs AAV9 MAAPΔ Total, ***$p = 0.0001$; AAV9 Total vs AAV9 MAAPΔ + MAAP8 Total, **$p = 0.0075$).

from cell culture media. Further, we note that mutation of the *cap* gene in engineered AAV capsids within the overlapping MAAP reading frame could impact vector yield and timing of harvest from cell lysate versus media fractions. Interestingly, we observed a significant titer increase on Day 3 with MAAP1Δ, MAAP8Δ, and MAAP9Δ vectors. One possible explanation for the increased titer is that in cells transfected with MAAPΔ plasmids, empty rAAV capsids are not as readily secreted into the media and are retained in the nucleus, thereby increasing their availability for REP mediated genome packaging. In addition, we also observed that MAAP contributes to rapid secretion of what appears to be a diverse population of EVs, independent of other AAV components. While this aspect needs to be explored further, it is exciting to note that MAAP might provide an orthogonal solution to loading therapeutic cargo onto EVs.

From a structural perspective, MAAP is rich in alpha helical domains, interspersed by T/S rich linker regions and basic residues. The most notable attribute of MAAP is a cationic amphipathic C-terminal domain, which is critical for membrane anchoring and extracellular secretion. Recent studies have revealed that PTGFRN and BASP1 can be utilized as scaffolds for loading various molecules onto the surface or into the lumen of EVs[22]. BASP1 is an inner membrane leaflet localized protein that utilizes a polybasic effector domain (PED) to bind to the EV membrane. Interestingly, the PED, along with N-terminal myristoylation of BASP1, is essential for proper loading of carrier proteins to the inner membrane of EVs. Similarly, the matrix (MA) membrane proximal domain of HIV-1 Pr55[GAG] contains a highly basic region that is important for efficient membrane binding and proper targeting of MA to the plasma membrane[32]. MAAP also contains multiple basic regions that are essential for proper molecular function; however, whether MAAP undergoes post-translational lipid modification remains to be determined. Nevertheless, our data suggests that similar to PTGFRN, MAAP can potentially be fused to protein cargo for loading onto EVs.

Furthermore, our work has shed light on the interaction between MAAP, AAV, and EVs. Using established MISEV criteria[33], we showed that a sub-population of secreted AAV particles associate with multiple EV markers (CD81, CD63 and CD9), while a fraction of AAV particles appears to be free of any vesicular association. Notably, the EV fractions showed strong MAAP association further implicating this viral protein in vesicle mediated AAV cellular egress. It is plausible that the free particles were previously associated with EVs that lysed during processing steps or MAAP alters the cell membrane through other mechanisms, which remain the subject of investigation. From a cell biology perspective, our data supports a model wherein MAAP hijacks the EV/exosomal pathway. Exactly how MAAP and AAV capsids work in tandem to utilize this pathway prior to secretion warrants further investigation. It is plausible that MAAP may enable passive loading of AAV particles into EVs during secretion or alternatively, actively recruit AAV capsids into EVs through an unknown cellular bridging factor. Indeed, studies exploring cellular egress of autonomous parvoviruses, such as Minute Virus of Mice (MVM), have shown viral particles to be sequestered into COPII-vesicles in the endoplasmic reticulum (ER) and transported via the Golgi compartment to the plasma membrane[34]. Remodeling of the cytoskeletal filaments through gelsolin mediated actin fiber degradation in infected cells have been shown to be essential for egress. Further, several biochemical factors such as SAR1, SEC24, RAB1, the ERM family proteins, radixin, and moesin have been implicated in this process of cellular exocytosis[35]. However, it should be noted that a virulence factor akin to MAAP in other parvoviruses has not been identified to date. In summary, we conclude that while dependoparvoviruses such as AAV rely on helper co-infection to trigger viral release through a lytic process, MAAP-mediated viral egress likely accelerates and regulates dissemination and spread. In summary, our studies unequivocally implicate MAAP as a novel AAV egress factor.

## Methods

**Plasmid constructs.** MAAP DNA sequences from AAV serotypes 1,2,5,8, and 9 were synthesized and cloned into pcDNA3.1(+)-C-HA and pcDNA3.1(+)-C-eGFP expression vectors using HindIII and XbaI sites for MAAP 1,2,8,9 and EcoRV sites for MAAP5 (Genscript). All MAAP expression constructs were synthesized and cloned with an ATG start codon. The AAV8-Rep/Cap-VP* plasmid is a AAV2-Rep/AAV8-Cap plasmid with the start codons of VP1/2/3 and AAP mutated by site-directed mutagenesis to prevent expression. The AAV8-Rep/Cap-MVP* additionally has a mutated MAAP start codon to prevent MAAP expression. The AAV8-MAAPΔ plasmid is a 2 AAV2-Rep/AAV8-Cap plasmid with the start codons of MAAP mutated by site-directed mutagenesis to prevent expression. The AAV8-MAAP-3X-FLAG plasmid was generated by utilizing site-directed mutagenesis to incorporate a 3X-FLAG tag in frame onto the C-terminus of MAAP in the AAV8-Rep/Cap-VP* plasmid. AAV8-MAAP-3X-FLAG truncation mutants were generated through site-directed mutagenesis. MAAP8 was cloned into the MCS-13X-Linker-BioID2-HA expression vector (Addgene #80899) using NheI and Age1 sites. All plasmid constructs were verified by DNA sequencing analysis.

**Bioinformatics analysis and structural models.** The amino acid sequences of 15 AAV serotypes were retrieved from GenBank. MAAP start and stop sites were defined as previously described[18]. Protein sequences were aligned using the ClustalW multiple-alignment tool[36] and generated using Unipro UGENE software[37]. MAAP amino acid sequences from multiple AAV isolates were aligned using ClustalW, and phylogenetic trees were generated using the MEGAv7.0.21 software package[38]. The phylogeny was produced using the neighbor-joining algorithm, and amino acid distances were calculated using a Poisson correction[39]. Statistical testing was done by bootstrapping with 1000 replicates to test the confidence of the phylogenetic analysis and to generate the original tree[40]. The percentage of replicate trees in which associated taxa clustered together in the bootstrap test is displayed next to the branches. Secondary structural elements were predicted using the JPred tool[41]. To predict membrane-binding, amphipathic α-helices, we used Amphipaseek (parameters: high specificity/low sensitivity)[42]. MAAP structural models were generated using the TrRosetta deep-learning-based modeling method[19]. Secondary structural depictions of these models were visualized using the PyMOL Molecular Graphics System (Schrödinger; https://www.pymol.org/2/).

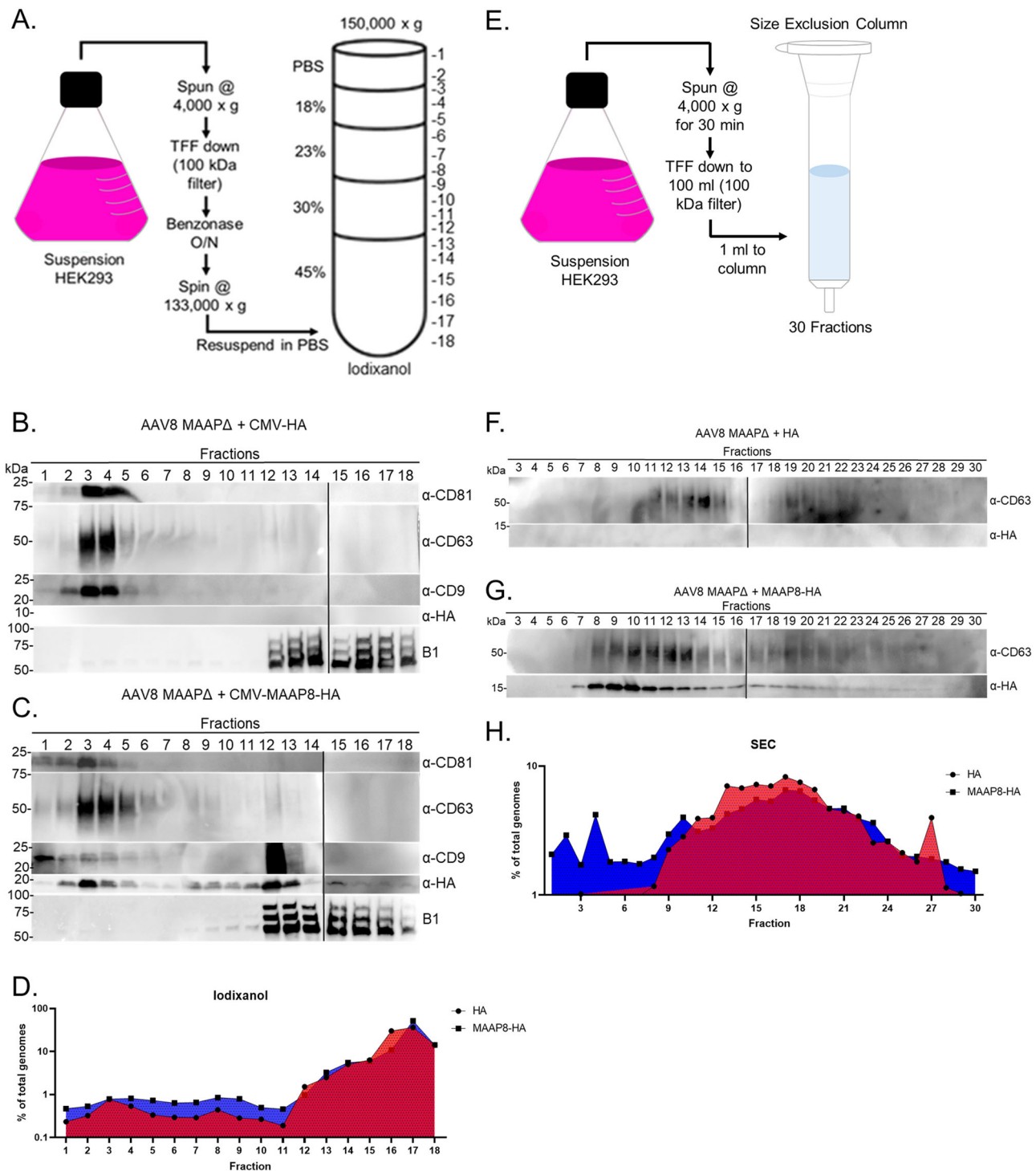

**Fig. 5 MAAP promotes association of AAV with EVs. A** Schematic of EV isolation by iodixanol density gradient from HEK293 suspension culture. **B** Immunoblots of iodixanol fractions from suspension cells producing recombinant MAAP8Δ vector complemented *in trans* with CMV-HA and **C** CMV-MAAP8-HA. EVs, capsid and MAAP were analyzed from the media of HEK293 producing cells at day 3 post transfection. Capsid and MAAP proteins were analyzed by SDS-PAGE under reducing conditions while EV markers (CD81, CD63, CD9) were analyzed by SDS-PAGE under non-reducing conditions (*n* = 2). **D** Graph displaying percent vector genome titer relative to total viral genomes for each fraction from iodixanol gradient purified MAAP8Δ vector complemented *in trans* with CMV-HA and CMV-MAAP8-HA. **E** Schematic of EV isolation by size exclusion chromatography (SEC) from HEK293 suspension culture. **F** Immunoblots of SEC fractions from suspension cells producing recombinant MAAP8Δ vector complemented *in trans* with CMV-HA and **G** CMV-MAAP8-HA. EVs and MAAP were analyzed from the media of HEK293 producing cells at day 3 post transfection. MAAP protein was analyzed by SDS-PAGE under reducing conditions while the EV marker CD63 was analyzed by SDS-PAGE under non-reducing conditions (*n* = 2). **H** Graph displaying percent vector genome titer relative to total viral genomes for each fraction from SEC purified MAAP8Δ vector complemented *in trans* with CMV-HA and CMV-MAAP8-HA.

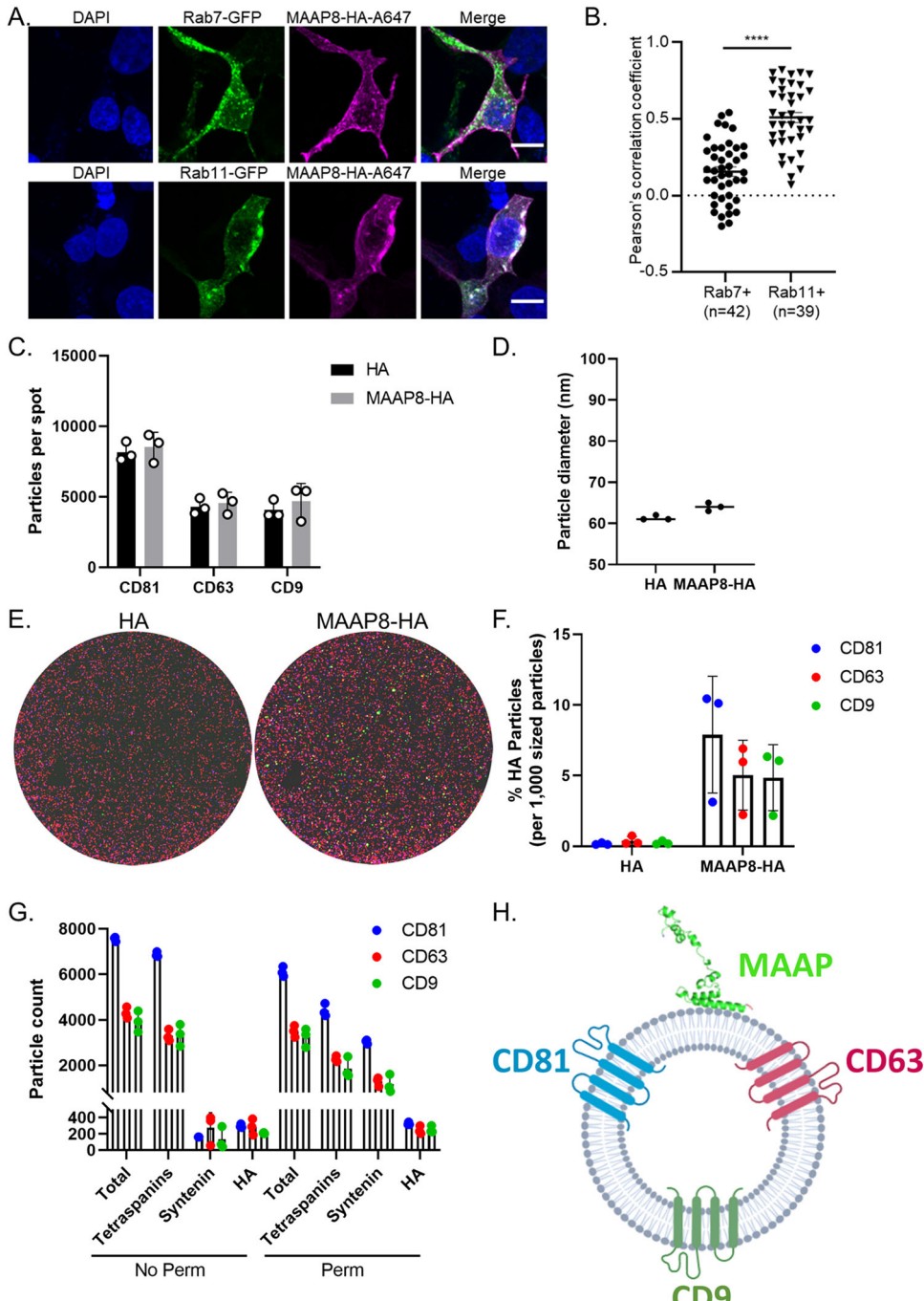

**Fig. 6 MAAP interacts with the surface of extracellular vesicles. A** HEK293 cells were transfected with expression vectors encoding Rab7-GFP, Rab11-GFP, MAAP8-HA. MAAP-HA was detected by immunofluorescence with an AlexaFlour647 secondary antibody (MAAP8-HA-A647). A Z-stack of confocal optical sections at 1-μm steps was acquired. A 3-μm-thick medial stack is shown. Images are representative of three experiments. Scale bars, 10 μm. **B** Co-localization between MAAP8-HA and Rab7-GFP or Rab11-GFP in the whole cell as assessed by Pearson's correlation coefficient (R) as described in Materials and Methods. Each dot represents one cell. Horizontal bars represent the mean ± SEM. A two-sided Mann–Whitney rank test was used to determine significance. (Rab7 + vs Rab11 + , ****$p < 0.0001$). **C** Chips coated with separate capture spots for anti-CD81, anti-CD63, and anti-CD9 were used to capture EVs from media of cells expressing CMV-HA and CMV-MAAP8-HA. Total captured EVs on each chip were quantified ($n = 3$). Data are presented as mean values ± SD. **D** Representative size distribution profile of tetraspanin-positive EVs determined by label-free interferometry ($n = 3$). **E** CD81 capture probe images of EVs captured from media of cells expressing CMV-HA and CMV-MAAP8-HA. EVs were probed with anti-CD81-CF488a (blue), anti-HA-Alexa Fluor 555 (green), anti-CD9-CF647 (red), and anti-CD63-CF647 (red) ($n = 3$). **F** Quantification of percent HA labeled EVs captured from media of cells expressing CMV-HA and CMV-MAAP8-HA ($n = 3$). Data are presented as mean values ± SD. **G** EVs from media of cells expressing CMV-MAAP8-HA were captured with anti-CD81, anti-CD63, and anti-CD9 coated chips. EVs were fixed, permeabilized and probed with anti-HA-Alexa Fluor 555 to determine the EV loading orientation of MAAP8-HA ($n = 3$). Data are presented as mean values ± SD. **H** Cartoon schematic depicting EV surface loading of MAAP.

**Cellular assays, immunoprecipitations, and western blotting**. For protein expression analysis, HEK293 cells seeded overnight in 6-well plates at a density of $3 \times 10^5$ cells per plate were transfected with a total of 2 µg DNA as indicated. HEK293 cell pellets overexpressing MAAP-HA, MAAP-GFP, or expressing MAAP8-3X-FLAG were recovered 72 h post transfection. Pellets were lysed in RIPA buffer with 1x Halt Protease Inhibitor (ThermoFisher) for 45 min at 4 °C. Lysates were spun at max speed for 10 min at 4 °C to remove cellular debris. 1x LDS sample buffer with 10 mM DTT were added to cleared lysates and boiled for 2 min. Samples of cleared lysate were ran on Mini-Protean TGX 4–15% gels (Biorad), transferred onto PVDF with the Trans-Blot Turbo system (BioRad) and blocked in 5% milk/1x TBST. Blots were probed with mouse monoclonal anti-GFP antibody (1:1000 dilution, SC9996; Santa Cruz Biotechnology), rabbit polyclonal anti-HA SG77 antibody (1:1000 dilution, 71-5500; ThermoFisher Scientific), mouse monoclonal B1 hybridoma supernatant (1:50, 03-65158; ARP), mouse monoclonal anti-FLAG M2 antibody (1:1000 dilution, F180450UG; Sigma), mouse monoclonal anti-beta-actin (1:1000 dilution, 8226; Abcam) as the primary antibody. Following three 1x TBST washes, samples were incubated with secondary antibodies conjugated to HRP at 1:20,000 in 5% milk/1x TBST for 1 hour (goat anti-mouse-HRP, 32430; ThermoFisher Scientific and goat anti-rabbit-HRP, 111-035-003; Jackson ImmunoResearch). Blots were developed using SuperSignal West Femto substrate (ThermoFisherScientific/Life Technologies) according to manufacturer instructions. For immunoprecipitation studies, HEK293 cells were transfected with pXR9 and MAAP9- pcDNA3.1(+)-C-HA for 72 h, then washed with 1x PBS and harvested in NP-40 with 1x Halt Protease Inhibitor (ThermoFisher) for 1 h at 4 °C. Lysates were spun at max speed for 20 min at 4 °C to remove cellular debris. In all, 10 µL (2.5 µg) of anti-HA SG77 antibody were added to 500 µL cleared lysate and incubated at 4 °C for 3 h with nutation. Added 40 µL of pre-washed Protein G magnetic beads to pre-cleared lysate with antibody and carried out immunoprecipitations overnight on a nutator at 4 °C. Bound protein was eluted in 10 mM DTT and 1x LDS for 5 min at 95 °C. Samples were then analyzed via SDS-PAGE (NuPAGE 4–12% Bis-Tris Gel) and transferred onto nitrocellulose membrane (ThermoScientific). Following blocking in 5% milk/1x TBST, samples were incubated with primary antibodies to either capsid (mouse monoclonal B1 hybridoma supernatant, 1:250, 65158; Progen), actin (mouse monoclonal anti-beta-actin, 1:1000 dilution, 8226; Abcam), CD81 (mouse monoclonal anti-CD81 M38, 1:1000, 10630D), CD63 (mouse monoclonal anti-CD63 Ts63, 1:1000, 10628D), CD9 (mouse monoclonal anti-CD9 Ts9, 1:1000, 10626D) or MAAP (mouse monoclonal anti-HA HA.C5 antibody, 1:1000 dilution, MA5-27543; ThermoFisher Scientific) overnight in 5% milk/1x TBST. Following three 1x TBST washes, samples were incubated with secondary anti-mouse antibody conjugated to HRP (goat anti-mouse-HRP, 32430; ThermoFisher Scientific) at 1:20,000 in 5% milk/1x TBST for 1 h. The signal was the visualized via SuperSignal West Femto Maximum Sensitivity substrate (ThermoScientific) according to manufacturer instructions.

**BioID2 expression and pulldown**. For protein expression analysis, HEK293 cells seeded overnight in 6-well plates at a density of $3 \times 10^5$ cells per plate were transfected with a total of 2 µg of 13X-BioID2-HA or MAAP8-13X-BioID2-HA DNA. Cells were then supplemented with 50 µM biotin 24 h post transfection and allowed to label for 20 h. Cell pellets were recovered 20 h biotin supplementation and were lysed in RIPA buffer with 1x Halt Protease Inhibitor (ThermoFisher) for 45 min at 4 °C. Lysates were spun at max speed for 10 min at 4 °C to remove cellular debris. 1X LDS sample buffer with 10 mM DTT were added to cleared lysates and boiled for 2 min. Samples of cleared lysate were ran on Mini-Protean TGX 4–15% gels (Biorad), transferred onto PVDF with the Trans-Blot Turbo system (BioRad) and blocked in 5% milk/1x TBST. Blots were probed with rabbit polyclonal anti-HA SG77 antibody (1:1000 dilution, 71-5500; ThermoFisher Scientific), mouse monoclonal anti-beta-actin (1:1000 dilution, 8226; Abcam), and goat polyclonal anti-biotin (1:1000 dilution, 31852; ThermoFisher) as the primary antibody. Following three 1x TBST washes, samples were incubated with secondary antibodies conjugated to HRP at 1:20,000 in 5% milk/1x TBST for 1 h (goat anti-mouse-HRP, 32430; ThermoFisher Scientific, goat anti-rabbit-HRP, 111-035-003; Jackson ImmunoResearch, mouse anti-goat-HRP, 31400; ThermoFisher).

Streptavidin pulldowns were performed as previously described[43] with the following changes. HEK293 cells for each condition were grown on $2 \times 15$ cm$^2$ plates until ~70% confluent and were transfected with 34 µg of plasmid DNA per plate. Each plate was transfected with 13X-BioID2 or MAAP8-13X-BioID2 along with pXX680, pTR-CBA-Luciferase, and AAV8-MAAPΔ. Media was supplemented with 50 µM biotin 48 h post transfection and cells were harvested 20 h post biotin supplementation. Biotinylated proteins were pulled down with Pierce High Capacity Streptavidin Agarose (20357; ThermoFisher) and were separated by SDS-PAGE and visualized using a SilverXpress silver stain kit (LC6100; ThermoFisher) according to the manufacturer's instructions. Biotinylated proteins were also visualized by immunoblot and probed with rabbit polyclonal anti-HA SG77 antibody (1:1000 dilution, 71-5500; ThermoFisher Scientific), mouse monoclonal anti-beta-actin (1:1000 dilution, 8226; Abcam), goat polyclonal anti-biotin (1:1000 dilution, 31852; ThermoFisher) and mouse monoclonal B1 hybridoma supernatant (1:50, 03-65158; ARP) as the primary antibody.

**Recombinant and wild type AAV production, purification, and quantification**. HEK293 (human embryonic kidney cells obtained from the University of North Carolina Vector Core) were maintained in Dulbecco's Modified Eagle's Medium (DMEM) supplemented with 10% fetal bovine serum (FBS), 100 U/mL penicillin, 100 µg/mL streptomycin. Cells were maintained in 5% CO$_2$ at 37 °C. Recombinant AAV vectors were produced by transfecting HEK293 cells at ~75% confluence with polyethylenimine (PEI Max; Polysciences, 24765) using a triple plasmid transfection protocol with the AAV Rep-Cap plasmid, Adenoviral helper plasmid (pXX680), and single-stranded genomes encoding firefly luciferase driven by the chicken beta-actin promoter (ssCBA-Luc) or self-complementary green fluorescence protein driven by a hybrid chicken beta-actin promoter (scCBh-GFP), flanked by AAV2 inverted terminal repeat (ITR) sequences. Viral vectors were harvested from media and purified via iodixanol density gradient ultracentrifugation followed by phosphate buffered saline (PBS) buffer exchange. Titers of purified virus preparations were determined by quantitative PCR using a Roche Lightcycler 480 (Roche Applied Sciences, Pleasanton, CA) with primers amplifying the AAV2 ITR regions (Supplementary Table 1).

**Quantitative PCR analysis of AAV vector yield**. HEK293 cells in six-well plates were transfected using PEI at ~75% confluence with Adenovirus helper plasmid (1 µg), WT or MAAPΔ AAV-Rep/Cap plasmid (1 µg), ITR-transgene plasmid (500 ng), and AAV8-Rep/Cap-VP* or -MVP* (500 ng). The AAV8-Rep/Cap-VP* plasmid is a AAV2-Rep/AAV8-Cap plasmid with the start codons of VP1/2/3 and AAP mutated to prevent expression. The AAV8-Rep/Cap-MVP* additionally has a mutated MAAP start codon. For 3-day experiments, media and cells were collected three days post transfection. Cells were lysed by vortexing in a mild lysis buffer (10 mM Tris-HCl, 10 mM MgCl2, 2 mM CaCl$_2$, 0.5% Triton X-100 supplemented with DNAse, RNAse, and Halt Protease Inhibitor Cocktail) and incubated at 37 °C for 1 h. Lysates were cleared by centrifuging at 21,000 rcf for 2 min. NaCl to 300 mM was added to AAV2 lysates prior to centrifugation to prevent virus binding of the cell debris. Collected media and cleared lysates were assayed with qPCR for DNAse-resistant viral genomes as described above. For Day 3 and 5 experiments, media was collected and replaced on the first indicated day, and cells and media were harvested as described on the last day. For WT-like AAV8 transfections, cells were transfected with Adenovirus helper plasmid (1.9 µg) and WT or MAAPΔ ITR- AAV2-Rep/AAV8-Cap -ITR plasmid (0.9 µg), with media and cells collected on days 3 and 5 post transfection as described above.

**Confocal fluorescence microscopy**. HEK293 cells were seeded on slide covers in 24-well plates at a density of 5e4 cells/well and allowed to adhere overnight. Cells were then co-transfected with Rab7-GFP, Rab11-GFP, MAAP8-pcDNA3.1(+)-C-HA and MAAP9-pcDNA3.1(+)-C-HA, and were incubated for 48 h at 37 °C and 5% CO$_2$. Cells were then fixed with 4% paraformaldehyde for 30 min and permeabilized with 0.1% Triton X-100 for 30 min. Following 1 h of blocking with 5% Normal Goat Serum, cells were stained with rabbit polyclonal anti-HA SG77 antibody (1:100 dilution, 71-5500; ThermoFisher Scientific) primary for one hour, washed 3x with PBS, and then stained with fluorescent goat anti-rabbit Alexa Fluor 647 (1:400 dilution, ab150079; Abcam) secondary antibody and washed 1x with PBS. Cells were subject to 5 min staining with DAPI, and then mounted in Prolong Diamond (Invitrogen) and imaged using a Zeiss LSM 880 Airyscan confocal microscope. Co-localization analysis was performed by cropping the whole compartment (Rab11 or RAB7), or the whole cell, using Zeiss ZEN software with the Co-localization function. Threshold was automatically determined using the Costes method autothreshold determination. Pearson's correlation coefficient was calculated for the analysis. Statistical analyses were carried out by the nonparametrical Mann–Whitney $U$ test using Prism software (GraphPad).

**Extracellular vesicle isolation (iodixanol and size exclusion chromatography)**. For Fig. 5A–D and Supplementary Fig. 4, a previously established EV isolation protocol was utilized[22]. Suspension adapted HEK293 (human embryonic kidney cells obtained from the University of North Carolina Vector Core) cells were grown in Dulbecco's Modified Eagle's Medium (DMEM) supplemented with 10% fetal bovine serum (FBS), 100 U/mL penicillin, 100 µg/mL streptomycin and inoculated at a density of 5E + 5 viable cells per milliliter. Cells were maintained in 5% CO$_2$ at 37 °C and grown overnight to a density of 1E + 6 cells per milliliter for transfection. Cells were then grown for 3 days with viability measured daily. Cell density and viability were measured on a Countess II Automated Cell Counter (Invitrogen). Upon culture termination, cells were removed by centrifugation at 4000 x $g$ for 30 min at room temperature. Harvests were then concentrated using tangential flow filtration using a Minikros (100 kD) modified polyethersulfone (PES) hollow fiber filter (Repligen) on a KROSFLO KMPI system. Concentrated medium was then supplemented with 1 mM MgCl$_2$ and benzonase (20 U/mL, Sigma Aldrich) to digest extravesicular nucleic acids. Nuclease treated media was next transferred to 38-mL Ultra-Clear centrifuge tubes (Beckman Coulter) and centrifuged for 60 min at 133,000 x $g$ at 4 °C in a Sure Spin 630 36 mL rotor (Thermo Scientific). The media was discarded and the crude pellet was resuspended in a minimal volume of PBS. The resuspended pellet was then mixed with a 60% iodixanol solution (Optiprep, Sigma) and layered onto the bottom of a 38-mL Ultra-Clear centrifuge tube (Beckman Coulter). Lower-density solutions were prepared by diluting homogenization buffer (250 mM sucrose, 10 mM Tris-HCl, 1 mM EDTA, pH 7.4) to yield final iodixanol concentrations (vol/vol) of 45%, 30%, 23%, and 18%. Successive layers of 30% (9 mL), 23% (6 mL), and 18% (6 mL)

iodixanol solutions were carefully pipetted on top. Resuspended pellet in PBS was added on top of the gradient. The gradient was ultracentrifuged in a Sure Spin 630 36 mL rotor (Thermo Scientific) for 16 h at 150,000 x $g$ and 4 °C to separate EVs from other cell culture supernatant contaminants. The gradient was then fractionated in 2 mL fractions and analyzed by biochemical analysis.

To purify EVs by size exclusion chromatography, CL-2B Sepharose resin (CL2B300, Sigma Aldrich) was washed with an equal volume of PBS in a glass container and placed at 4 °C to let the resin settle completely. PBS washes were repeated two more times for a total of three washes. Columns were prepared fresh on the day of use. Washed resin was poured into an Econo-Pac Chromotography column (Bio-Rad, 7321010) and the bed volume brought to 10 mL. The top frit was immediately placed at the top of the resin and the column was subsequently washed with 20 mL PBS. Immediately before sample addition, the column was allowed to fully drip out and 1 mL of sample was added to the column. As soon as sample was added to the top of the column, PBS was added to the top of the column 1 mL at a time. Fraction numbers corresponded to 0.5 mL increments collected as soon as sample was added. Thirty total fractions were collected and analyzed by SDS-PAGE and western blot.

**Single-particle interferometric reflectance imaging sensing (NanoView) analysis**. Samples were processed as described previously[44–47]. Samples were diluted according to the manufacturer's protocol for the ExoView tetraspanin kit (NanoView Biosciences; EV-TETRA) and incubated for 16 h on ExoView tetraspanin chips spotted with antibodies against CD81, CD63, CD9 and the mouse IgG1,κ isotype control in triplicate. The chips were then washed in an automated chip washer and incubated with conjugated antibodies for fluorescent labeling of the captured EVs (anti-CD81- CF488a, anti-HA-Alexa Fluor 555 [AF555] (ThermoFisher, 26183-A555), anti-CD9-CF647, and anti-CD63-CF647 for 1 h. After labeling, the chips were washed and dried in the automated chip washer and placed in the reader for analysis. All data were gathered using an ExoView R100 reader equipped with ExoView Scanner 3.0 software and analyzed using ExoView Analyzer 3.0.

**Transmission electron microscopy**. Isolated vesicular fractions or AAV viral particles ($1 \times 10^{10}$ vg) in 1x PBS samples were adsorbed onto 400 mesh, carbon coated grids (Electron Microscopy Sciences) for 2 min and briefly stained with 1% uranyl acetate (Electron Microscopy Sciences) diluted in 50% ethanol. After drying, grids were imaged with a Philips CM12 electron microscope operated at 80 kV. Images were collected on an AMT camera.

**Luciferase expression assays**. A total of $1 \times 10^4$ HEK293 cells in 50 µL DMEM + 10% FBS + penicillin streptomycin was then added to each well, and the plates were incubated in 5% CO2 at 37 °C for 24 h. Cells were then transduced with AAV8-ssCBA-Luc vectors at a dose of 10,000 and 50,000 vg/cell. Forty-eight hours post transduction, cells were then harvested and lysed with 25 µL of 1 × passive lysis buffer (Promega) for 30 min at room temperature. Luciferase activity was measured on a Victor 3 multi-label plate reader (PerkinElmer) immediately after the addition of 25 µL of luciferin (Promega).

**Statistical analysis**. Where appropriate, data are represented as mean or mean ± standard deviation. For data sets with two groups, comparisons were made between all groups and significance was determined using a two-way ANOVA followed by a Sidak's post- test. For data sets with at least three groups, comparisons were made between all groups and significance was determined by two-way ANOVA, with Tukey's post-test. For analysis of confocal microscopy data (Fig. 6A, B and Supplementary Fig. 6A, B) significance was determined by a Mann–Whitney rank test. *$p < 0.05$, **$p < 0.01$, ***$p < 0.001$, ****$p < 0.0001$.

## Data availability

The data that support the findings of this study are available from the corresponding author upon request. Source data are provided with this paper.

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

## Acknowledgements

This study was funded by NIH grants awarded to A.A. (R01HL089221, UG3AR07336, R01GM127708; R01NS099371). We would like to acknowledge Ricardo Vancini and the Duke Research Electron Microscopy Service for their help in the preparation and imaging of TEM samples. We would also like to acknowledge Dana Elmore for her assistance with data analysis.

## Author contributions

Z.E., P.H. and A.A. designed all experiments and interpreted the data. Z.E., P.H. and D.O. carried out all molecular biology, virus production, and imaging studies. L.A. and G.D. performed single-particle interferometric reflectance imaging sensing (NanoView) analysis of exosomes. Z.E., P.H., H.V. and A.A. wrote the manuscript.

## Competing interests

A.A. and Z.E. have filed patent applications on the subject matter of this manuscript. A.A. is a co-founder at StrideBio and TorqueBio and an advisor to Sarepta Therapeutics, Mammoth Biosciences, Atsena Therapeutics, Ring Therapeutics, AstraZeneca Pharmaceuticals. A.A. and Z.E. are advisors to Isolere Bio.
