## [Peer Review File · Nature Communications]

Reviewer comments, initial round of review: - -

Reviewer #1 (Remarks to the Author):

The manuscript by Dr. Asokan and colleagues conducted a thorough research on MAAP, a novel protein encoded in +1 ORF of cap gene. After Ogden et al. reported MAAP in 2019, this is the first paper not only confirmed the existence of MAAP also reported many crucial aspects on the role of MAAP in AAV virology. The data shows that like AAP, MAAP can be trans-complemented to mismatched AAV serotypes and regained the secretion kinetics. Also, MAAP Δ from AAV1, AAV8 and AAV9 seems to significantly increase the overall titer from WT, which could potentially benefit the AAV production. By determining MAAP protein domains, it provides a foundation of understanding and engineering MAAP for the purpose of increasing AAV titer.

This reviewer feels that the majority of the experimental studies and reporting meets high standards. Particularly, I'm impressed that authors were able to clearly separate exosome and microvesicle pathways and identified that MAAP is associated with the exosomal pathways of cells, which could be easily confused.

Overall all, the manuscript is well written, flows logically, and the data is communicated concisely and clearly

However, there are a several areas of the manuscript that require further clarification and/or explanation. These minor points are outlined below:

- 1) In p.5, author mentioned "MAAP5 and 9 showed significant divergence from other serotypes." However, it should be MAAP5 and 2 instead based on the phylogenic tree.
- 2) In figure 2, author compared that titers from day 3 and day 5. However, did authors look into the kinetics within the day 1 or day 2 as it may be more relevant to the conventional AAV production protocol. Additionally, cells are not healthy by day 5. The titer data from day 1 and day 2 from media, cells and total may be more informative.
- 3) Please add the statistical analysis for supplementary Fig. 2A.
- 4) Please specify the genome packaged in AAVs in Fig. 4.
- 5) For the data in Fig. 5G and 5H, could authors comment on if immnogold labeling of tagging AAVs would be more directly show that MAAP mutant affects the location of AAV in exosome?
- 6) Although author did not stress on this, MAAP1 Δ , MAAP8 Δ and MAAP9 Δ on day 3 all show significantly higher rAAV titers than those of their respective wt forms. Could authors discuss the following possibilities: 1) MAAP is toxic to cells, 2) less cell death in MAAP Δ groups? 3). If MAAP is not toxic to cells, what potential mechanism to explain the higher titer in MAAP Δ ?
- 7) Figure 5H legend is missing.

Reviewer #2 (Remarks to the Author):

Recommendation: Strong accept

Elmore et al. investigate the function of the recently discovered AAV MAAP protein. AAV capsids are an extremely important component of emerging gene therapies, yet many aspects of AAV basic biology are poorly understood, including the role of MAAP in the natural AAV lifecycle and in the recombinant AAV systems commonly applied for gene therapy. Through a series of elegant experiments and thoughtful genetic controls, the authors show that MAAP facilitates viral egress from cells. These results provide the first clear insights into the functional role of MAAP and also help to explain the differences in cell retention and secretion observed among natural AAV capsid serotypes. The authors go on to show that MAAP expressed in trans of the viral genome can enhance the secretion of recombinant AAV particles, which has immediate potential applications

towards improving purification processes of AAV gene therapies. In answering key open questions for the AAV field through mechanistic insights and through direct demonstration of practical applications, this study would be of significant interest to biochemists, virologists, translational researchers and protein engineers.

The experiments were competently designed and the manuscript is clearly organized and written. I recommend acceptance and have only a few small comments that the authors may consider if they wish to improve the clarity and interpretability of their results.

-Line 155: The authors state that deletions of the N or C terminus severely diminished MAAP expression. This is evident from looking at the MAAP8_deltaN and MAAP8_deltaC bands in Fig 3b. However it is confusing that MAAP8_delta1-2, MAAP8_delta1-3, MAAP8_delta1-4 and MAAP8_deltaN1 are shown as bands at high levels of expression in Figs 3bc. In other words, it is not clear from the manuscript or from Fig 3a how MAAP8_deltaN differs from MAAP8_delta1 and the related mutants that also contain a deletion of region 1. The authors should provide more documentation for sequences of the exact constructs used in the structure-function analysis, as well we clarify this information in the manuscript so that a reader can parse the figures without needing to look into the supplementary info.

Reviewer #3 (Remarks to the Author):

In their paper the authors study the viral membrane-associated accessory protein (MAAP) in promoting EV-mediated viral egress of AAV from host cells. They claim that secondary structure elements in MAAP contribute to its secretory function and demonstrate that MAAP significantly alter the kinetics of viral secretion of multiple AAV serotypes. Notably independent of AAV infection, MAAP alone may act as a stimulator of EV production and/or release.

This is a nicely conducted study with several interesting findings when AAV biology is concerned. The authors show convincingly that viral egress is critically dependent on the capsid protein MAAP and specifically it's C-terminal domain that holds a membrane binding, cationic amphipathic peptide (residues 96-114). Unfortunately, the claim that MAAP supports EV-mediated AAV egress is not convincingly demonstrated.

Major points;

The authors use one cell system (HEK293) that predominantly produces EVs from the PM. Hence the term exosomes should be avoided as in the EV-field such EVs are derived from internal compartments (MVBs) and are released through fusion of MVBs with the PM. However small EVs can also bud from the PM. Unfortunately the mechanism of egress is not studied in this paper.

In fact, the authors isolate (precipitate) their EVs with Exoquick (after 72hours!). This method does not discriminate between soluble protein complexes, EVs from the PM or internal compartments leading to uninterpretable results. EVs/Exosomes should be characterized according to MISEV criteria. A consequence of their choice to use exoquick

is that their EM images are not convincing. There is little proof the AAV are released via a vesiculation pathways, ideally KO cells should be used to delineate these for example KO for RAB27a/b, ESCRTs etc could/should be considered. Also, there is no data on the topology of the protein in the membranes. A MAAP-specific Ab should be made-used to study endogenous localization and trafficking. Indeed, the immunofluorescent data in fig 4 is not convincing as to where the protein is located, higher resolution is needed.

Overall, their data shows that MAAP is critically important for viral egress from HEK293 cells but the mechanism(s) and involvement of EV pathways is not supported by their data.

MISEV

<https://www.tandfonline.com/doi/full/10.1080/20013078.2018.1535750>

We thank the reviewers for their kind comments highlighting our study as “a series of elegant experiments and thoughtful genetic controls” and “answering key open questions for the AAV field through mechanistic insights and through direct demonstration of practical applications, that “would be of significant interest to biochemists, virologists, translational researchers and protein engineers.” We now provide a point by point response to reviewer comments focused on helping us improve the overall manuscript as well as highlight additional experiments and new data that corroborate a novel mechanism of AAV cellular egress.

Response to Reviewer 1

1. In p.5, author mentioned “MAAP5 and 9 showed significant divergence from other serotypes.” However, it should be MAAP5 and 2 instead based on the phylogenetic tree.

We appreciate the reviewer’s observation and comment. The manuscript now reads “MAAP2, 5 and 9 showed significant divergence from other serotypes”.

2. In figure 2, author compared that titers from day 3 and day 5. However, did authors look into the kinetics within the day 1 or day 2 as it may be more relevant to the conventional AAV production protocol. Additionally, cells are not healthy by day 5. The titer data from day 1 and day 2 from media, cells and total may be more informative.

We thank the reviewer for this question. From a suspension culture perspective, the current standard for harvesting AAV from HEK293 lysate is typically Day 3-5 (e.g., Piras et al., 2016). Nevertheless, we have now carried out analysis of titers on Days 1-2 from media as well as cell lysate as recommended (see attached data showing titers and western blots for Days 1 and 2). We observed that titers were not appreciably above background (potentially due to interference from transfected plasmid DNA). However, we do recover capsid protein secreted into media as determined by western blot analysis. The data is presented below. In general, our conclusion is that AAV secretion at earlier time intervals is not readily determined.

3. Please add the statistical analysis for supplementary Fig. 2A.

Statistical analysis has now been added to Supplementary Fig. 2A along with additions to the figure legend.

4. Please specify the genome packaged in AAVs in Fig. 4.

The genome (sc-CBh-GFP) has been added to the figure legend of figure 4.

5. For the data in Fig. 5G and 5H, could authors comment on if immunogold labeling of tagging AAVs would be more directly show that MAAP mutant affects the location of AAV in exosome? *This is an interesting suggestion however this is a difficult experiment since immunogold typically has a hydrodynamic diameter of 10-100 nm which may not be able to access the capsid without disrupting vesicles. While this experiment might be feasible for immunogold EM of fixed cells, this presents a significant technical challenge particularly for HEK293 cells. We feel strongly that the conclusions from Figure 5 validate the colocalization of AAV capsids with multiple extracellular vesicle (EV) markers within the scope of the current study.*

6. Although author did not stress on this, MAAP1 Δ , MAAP8 Δ and MAAP9 Δ on day 3 all show significantly higher rAAV titers than those of their respective wt forms. Could authors discuss the following possibilities: 1) MAAP is toxic to cells, 2) less cell death in MAAP Δ groups? 3). If MAAP is not toxic to cells, what potential mechanism to explain the higher titer in MAAP Δ ? *We thank the reviewer for this astute observation. One possible explanation for the increased titer is that in cells transfected with MAAP Δ virus, empty rAAV capsids are not as readily secreted into the media and are retained in the cells thereby increasing the possibility of REP mediated genome packaging. While we have not observed decreased cell death under MAAP Δ conditions, we have observed cellular toxicity when overexpressing MAAP8 with a strong promoter lending credence to the possible mechanism highlighted by the reviewer. We have highlighted the increased titers and possible mechanisms (lines 268-273 discussion section).*

7. Figure 5H legend is missing.

The legend has been added to the updated manuscript.

Response to Reviewer 2

1. Line 155: The authors state that deletions of the N or C terminus severely diminished MAAP expression. This is evident from looking at the MAAP8_deltaN and MAAP8_deltaC bands in Fig 3b. However it is confusing that MAAP8_delta1-2, MAAP8_delta1-3, MAAP8_delta1-4 and MAAP8_deltaN1 are shown as bands at high levels of expression in Figs 3bc. In other words, it is not clear from the manuscript or from Fig 3a how MAAP8_deltaN differs from MAAP8_delta1 and the related mutants that also contain a deletion of region 1. The authors should provide more documentation for sequences of the exact constructs used in the structure-function analysis, as well we clarify this information in the manuscript so that a reader can parse the figures without needing to look into the supplementary info.

We apologize for the confusion. As recommended, we have now replaced the schematic representation of MAAP mutants with a sequence alignment clearly listing all amino acid deletions in Figure 3.

Response to Reviewer 3

1. The authors use one cell system (HEK293) that predominantly produces EVs from the PM. Hence the term exosomes should be avoided as in the EV-field such EVs are derived from internal compartments (MVBs) and are released through fusion of MVBs with the PM. However small EVs can also bud from the PM. Unfortunately the mechanism of egress is not studied in this paper.

We have taken this critique into consideration and acknowledge that the term exosomes in the context of HEK293 cells producing AAV should be replaced with extracellular vesicles (EVs). Accordingly, we have now incorporated this recommended change throughout the manuscript. However, we do want to reiterate that the current study is primarily focused on a viral component that dictates cellular egress and as such a detailed investigation of cell biology mechanisms is outside the scope of this study and remains a topic of ongoing exploration.

In fact, the authors isolate (precipitate) their EVs with Exoquick (after 72hours!). This method does not discriminate between soluble protein complexes, EVs from the PM or internal compartments leading to uninterpretable results. EVs/Exosomes should be characterized according to MISEV criteria. A consequence of their choice to use exoquick is that their EM images are not convincing. There is little proof the AAV are released via a vesiculation pathways, ideally KO cells should be used to delineate these for example KO for RAB27a/b, ESCRTs etc could/should be considered. Also, there is no data on the topology of the protein in the membranes. A MAAP-specific Ab should be made-used to study endogenous localization and trafficking. Indeed, the immunofluorescent data in fig 4 is not convincing as to where the protein is located, higher resolution is needed. Overall, their data shows that MAAP is critically important for viral egress from HEK293 cells but the mechanism(s) and involvement of EV pathways is not supported by their data.

We have carefully thought through these comments and carried out additional experiments to characterize the secreted AAV using MISEV criteria (New Figure 5, Supplementary Figure 7). First, we would like to point out that over 20 published exosome/EV focused studies cite the ExoQuick kit. Therefore, we feel that the use of the kit to characterize secreted AAV particles in the context of exosomes/EVs is justified. Nevertheless, we have taken the reviewer's suggestion into consideration and utilized a thorough ultracentrifugation based exosome/EV purification method (Maguire et al., 2012; Dooley et al., 2021) to analyze secreted AAV fractions. This data revealed AAV particles that are associated with an extracellular vesicular (EV) fraction characterized by multiple markers (CD81, CD63 and CD9, generally acknowledged to be exosomal). We also observed free AAV in non-vesicular marker fractions. Notably the EV fractions in particular showed strong MAAP association implicating this viral protein in AAV cellular egress. Whether the free particles were previously associated with vesicles that lysed or were secreted independently remains the subject of investigation. As indicated earlier, we have shown strong association of MAAP with the Rab11 pathway; additional cell biology experiments investigating the ESCRT pathway are outside the scope of the current study. Most importantly as recommended by the reviewer, we have also included a discussion interpreting the results using guidelines set by MISEV criteria. We feel that the body of data presented in this study provides a roadmap for in depth cell biology experiments in the future. Overall, our study unequivocally confirms that MAAP is an adeno-associated viral egress factor.

Reviewer comments, second round of review: - -

Reviewer #1 (Remarks to the Author):

the authors have addressed this reviewer's comments point-by-point very well. This reviewer is satisfied with the revised version of the manuscript.

Reviewer #2 (Remarks to the Author):

The authors have sufficiently addressed my prior comments

Reviewer #3 (Remarks to the Author):

The authors have addressed some concerns but certainly not all. Their argument that 20 papers have been published using exoquick reagent for EV studies does not necessarily justify the use (or accurateness) of this reagent for all EV purposes. Indeed Exoquick precipitates all type of particles from biofluids including virions and EVs. This reagent can however be used in combination with other techniques, like density gradients or size-based separation. Indeed my main remaining issue concerns this, that it is still unclear whether MAAP truly induces EV production and or release and is not simply sorted into EVs that are secreted anyways. I do not find suppl figures 7 and 8 very convincing to make that claim.

Indeed what can be observed from the density gradient study that MAAP seems present in EV-enriched density fractions but also in what we suspect are non-EV fractions, the authors conclude; "although a fraction of AAV particles appears to be free of any vesicular association." However are these really particles or not soluble protein and/or protein complexes? In other words i feel AAV particles and EVs have not been convincingly separated.

Moreover where is the quantitation that MAAP induces EV-release and in parallel AAV egress?

What is missing is a separation based on size. Only taking size and density into account one may separate MAAP+ EVs from other (virus-like) particles, virions and protein complexes.

I suggest the authors use size exclusion chromatography (SEC) widely used in the field, to show MAAP and EV proteins in EV-sized fractions. It would behoove the authors if they perform both (DOI: 10.1038/s41596-019-0236-5), but i can accept a SEC-only experiment to complement their nice new density gradient data.

This is important as i have a hard time distinguishing EVs from viral capsids. The EVs the authors show in the supplementary data 8 look very small (<50nm) which approaches the size of viral capsids. I'm convinced that MAAP is released through EVs but the authors claim a role for RAB11 but only show in my opinion rather poor co-localization data, I suggest to use a siRNA or shRNA knockdown for RAB11 in AAV producing or MAAP expressing HEK cells and measure MAAP release in EV fractions.

Finally a better quantitation of EV secretion from HEK cells either producing the virus or with MAAP alone should be done. The westerns in suppl fig 7 and 8 are not sufficient, the authors can use NTA (nanoparticle tracking analysis device) to show increase in EV release, or use western blotting with proper normalization of protein expression in Cells and EVs. Tetraspanins alone are not sufficient, the authors may use TSG101, ALIX, HSP70, syntenin or other proteins to quantify EV production.

REVIEWER COMMENTS

Reviewer #1 (Remarks to the Author):

the authors have addressed this reviewer's comments point-by-point very well. This reviewer is satisfied with the revised version of the manuscript.

Reviewer #2 (Remarks to the Author):

The authors have sufficiently addressed my prior comments

We would like to thank Reviewers 1 and 2 for their continued support of this contribution.

Reviewer #3 (Remarks to the Author):

The authors have addressed some concerns but certainly not all. Their argument that 20 papers have been published using exoquick reagent for EV studies does not necessarily justify the use (or accurateness) of this reagent for all EV purposes. Indeed Exoquick precipitates all type of particles from biofluids including virions and EVs. This reagent can however be used in combination with other techniques, like density gradients or size-based separation. Indeed my main remaining issue concerns this, that it is still unclear whether MAAP truly induces EV production and or release and is not simply sorted into EVs that are secreted anyways. I do not find suppl figures 7 and 8 very convincing to make that claim.

We have now taken this critique into consideration and included both SEC based purification of different fractions as well as NanoView EV analysis to demonstrate that MAAP is associated with different vesicular populations. Moreover, MAAP also induces a significant shift in AAV-associated vesicular populations as corroborated by qPCR of viral genomes and an EV marker. Figures 5 and 6 highlight these findings. We elaborated further on these nuances in the discussion section.

Indeed what can be observed from the density gradient study that MAAP seems present in EV-enriched density fractions but also in what we suspect are non-EV fractions, the authors conclude; "although a fraction of AAV particles appears to be free of any vesicular association." However are these really particles or not soluble protein and/or protein complexes? In other words i feel AAV particles and EVs have not been convincingly separated. Moreover where is the quantitation that MAAP induces EV-release and in parralell AAV egress?

We would like to point out that we never indicated that MAAP/EVs were the only mechanism involved. It is likely that multiple, redundant mechanisms for AAV cellular egress are enabled by MAAP and this is the key highlight of the manuscript. Our data corroborates that AAV capsid protected genomes in fact associate with different vesicular subpopulations that co-elute in earlier CD63+ fractions. This is likely the same high density population visualized in the iodixanol gradient.

What is missing is a separation based on size. Only taking size and density into account one may

separate MAAP+ EVs from other (virus-like) particles, virions and protein complexes. I suggest the authors use size exclusion chromatography (SEC) widely used in the field, to show MAAP and EV proteins in EV-sized fractions. It would behoove the authors if they perform both (DOI: 10.1038/s41596-019-0236-5), but i can accept a SEC-only experiment to complement their nice new density gradient data.

These experiments have now been executed and we have gone a step further and carried out NanoView EV analysis as well to complement our data.

This is important as i have a hard time distinguishing EVs from viral capsids. The EVs the authors show in the supplementary data 8 look very small (<50nm) which approaches the size of viral capsids. Im convinced that MAAP is released through EVs but the authors claim a role for RAB11 but only show in my opinion rather poor co-localization data, I suggest to use a siRNA or shRNA knockdown for RAB11 in AAV producing or MAAP expressing HEK cells and measure MAAP release in EV fractions.

We feel strongly that mechanistic studies probing involvement of subcellular trafficking pathways is outside the scope of the current study. Our intent is to keep the focus on MAAP as a novel viral egress factor.

Finally a better quantitation of EV secretion from HEK cells either producing the virus or with MAAP alone should be done. The westerns in suppl fig 7 and 8 are not sufficient, the authors can use NTA (nanoparticle tracking analysis device) to show increase in EV release, or use western blotting with proper normalization of protein expression in Cells and EVs. Tetraspanins alone are not sufficient, the authors may use TSG101, ALIX, HSP70, syntaxin or other proteins to quantify EV production.

These experiments have now been executed. The NanoView EV analysis data convincingly demonstrate that MAAP is associated with the EV exterior (see Figure 6).

Reviewer comments, third round of review: - -

Reviewer #3 (Remarks to the Author):

I feel the authors have successfully and convincingly dealt with my remaining concerns and congratulate them with a beautiful paper that should be published in Nature Communications.